# An in-depth examination of the fuzzy fractional cancer tumor model and its numerical solution by implicit finite difference method

**Hamzeh Zureigat**[1¤*], **Saleh Alshammari**[2], **Mohammad Alshammari**[2], **Mohammed Al-Smadi**[3,4], **M. Mossa Al-Sawallah**[2]

**1** Faculty of Science and Technology, Department of Mathematics, Jadara University, Irbid, Jordan, **2** Department of Mathematics, College of Science, University of Ha´il, Ha´il, Saudi Arabia, **3** College of Commerce and Business, Lusail University, Lusail, Qatar, **4** Nonlinear Dynamics Research Center (NDRC), Ajman University, Ajman, UAE

¤ Current address: Applied Science Research Center, Applied Science Private University, Amman, Jordan
* hamzeh.zu@jadara.edu

## Abstract

The cancer tumor model serves a s a crucial instrument for understanding the behavior of different cancer tumors. Researchers have employed fractional differential equations to describe these models. In the context of time fractional cancer tumor models, there's a need to introduce fuzzy quantities instead of crisp quantities to accommodate the inherent uncertainty and imprecision in this model, giving rise to a formulation known as fuzzy time fractional cancer tumor models. In this study, we have developed an implicit finite difference method to solve a fuzzy time-fractional cancer tumor model. Instead of utilizing classical time derivatives in fuzzy cancer models, we have examined the effect of employing fuzzy time-fractional derivatives. To assess the stability of our proposed model, we applied the von Neumann method, considering the cancer cell killing rate as time-dependent and utilizing Caputo's derivative for the time-fractional derivative. Additionally, we conducted various numerical experiments to assess the viability of this new approach and explore relevant aspects. Furthermore, our study identified specific needs in researching the cancer tumor model with fuzzy fractional derivative, aiming to enhance our inclusive understanding of tumor behavior by considering diverse fuzzy cases for the model's initial conditions. It was found that the presented approach provides the ability to encompass all scenarios for the fuzzy time fractional cancer tumor model and handle all potential cases specifically focusing on scenarios where the net cell-killing rate is time-dependent.

## 1. Introduction

Fractional partial differential equations serve as essential tools for modeling various medical phenomena, with one prominent example being the study of cancer tumors. Cancer manifests

**Data Availability Statement:** All relevant data are within the manuscript and its Supporting information files.

**Funding:** This research has been funded by scientific research deanship at university of Ha'il -Saudi Arabia through project numemper (RG-23122).

**Competing interests:** NO authors have competing interests.

as the uncontrollable proliferation and dissemination of abnormal cells throughout the body. It encompasses a range of tumor types, including brain tumors, lung tumors, bone tumors, pancreatic tumors, organ tumors and others. Consequently, the lack of understanding cancer tumors has spurred comprehensive research in this area, drawing the interest not only of Bio-chemical and medical scientists but also of mathematicians.

Numerous methods have been introduced to investigate the development and responses to treatment of cancerous tumors. Many of these methods utilize statistical models, including the experimental methodologies or expectation-maximization technique [1–4]. In these investiga-tions, tumor growth or regression is analyzed as a function of time. Recently, Laajala et al. [2] introduced a statistical approach to replicate the evolution of tumor cells over time using time-dependent functions. Also, Benzekry et al. [1] proposed a Prototype to examine the prolifera-tion of cancer cells, incorporating a growth equation with varying constant rates. Additionally, Burgess et al. [5] put forward a diffusion-based prototype model to elucidate the interplay between growth rates when dealing with spherical cancer tumors that exhibit proliferation rates (p) and therapy-dependent killing rates (k). After that, Leach and Moyo [6] investigated the form of this Prototype pattern by utilizing Lie symmetry approach with independent vari-ables killing rate where, the rate at which cancer tumor cells are being eliminated, denoted as 'K,' is described as potentially varying with both time and position, rather than being solely dependent on constants or time alone, as suggested by the main hypothesis.

In recent decades, there has been increasing interest in fractional differential equations, leading to their application in explaining and developing models for actual occurrences in the real world that cannot be adequately addressed via classical PDEs. The substantial impact of fractional derivatives has provided an even more detailed portrayal of specific real-world issues. In the context of cancer tumor dynamics, several crucial characteristics need to be con-sidered when mathematically modeling tumor growth and progression. These include irregu-lar and complex patterns and shapes, non-linear growth dynamics, variations in cell distribution across various temporal and scales of spatial. These characteristics can be mea-sured utilizing fractal dimensions and different fractional derivatives of variables and parame-ters, which, in turn, facilitate enhanced comprehension of how tumor growth, as well as the potential for enhanced treatment and diagnosis planning. Many researchers have discussed these issues in detail in recent literature, with references ranging from [7–10].

Iomin introduced an important application of fractional derivatives in the context of cancer tumor modeling (referenced as [11]), specifically focusing on the super diffusion of cancer within a comb-like structure. This research demonstrates that the growth of the tumor is in agreement with the concept of fractional cell transport. This finding extends the understand-ing of fractional transport, particularly when addressing the question of how cancer cells can appear at distant locations originating from the primary tumor in instances of solid tumors. The paper presented an exact solution to this proposed model. Concurrently, Iyiola and Zaman, in their work [3], presented a cancer tumor prototype that utilizes time-fractional derivatives. They discussed the significance of using fractional order derivatives as opposed to conventional first-order integer time derivatives. The authors explored three distinct scenarios for the killing rate. For first scenario, the net rate of killing cells is solely time-dependent. While the second scenario, it is solely dependent on spatial factors. In the third scenario, it is contingent on the concentration or density of tumor of cancer cells. The research also revealed that a fractional derivative in time with an order $\alpha$ in the range [0,1] is a suitable prototype for the first scenario, while for the third scenario, a fractional derivative in time with an order $\alpha$ in the range *1,2] was advised as a more effective approach.

In actuality, real-world phenomena are often ambiguous and involve uncertainties and ambiguous in the values of the governing prototype quantities. This uncertainty, referred to as

uncertainty, is prevalent in various fields such as medicine, manufacturing, and engineering [12–15]. Fuzziness can emerge during data collection and measurement, as well as when determining initial and boundary conditions. In the context of diffusion equations with time fractional derivative, one can introduce fuzzy quantities instead of precise ones to accommodate the inherent uncertainty and imprecision, giving rise to a formulation known as a fuzzy fractional diffusion equation. Researchers have noted that cancer tumor models, represented by fractional diffusion equations [16–29], also consist of uncertain crisp quantities. Consequently, utilizing a fuzzy cancer tumor model (FCTM) becomes necessary to address this uncertainty. A recent study by Keshavarz et al. [30] tackled the FCTM utilizing a specific analytical approach. The researchers solved the FCTM under Caputo Hukuhara partial differentiability. This approach enabled a discussion on the effect of the fuzzy net cell-killing rate within the tumor, enhancing the comprehension of the model.

In the context of time fractional cancer tumor models, there's a need to introduce fuzzy quantities instead of crisp quantities to accommodate the inherent uncertainty and imprecision in this model, giving rise to a formulation known as fuzzy time fractional cancer tumor models. Therfore, to better understand the evolution of cancer tumors, it is valuable to explore the model under various fuzzy scenarios and fractional derivative conditions, as the net cell-killing rate plays a crucial role in monitoring tumor growth or regression. This investigation could assist researchers in selecting specific treatment strategies and offering a more comprehensive and practical depiction of cancer tumor behavior.

Zureigat et al. (2023) [40] developed and applied an explicit method to solve the fuzzy time-fractional cancer tumor model. They examined the impact of using the fuzzy time-fractional derivative rather than classical time derivatives in cancer tumor models. Additionally, they discussed the stability of the explicit method and found that the system is conditionally stable, indicating its inability to encompass all scenarios. This is evident in the article, where the authors restrict their discussion to a few specific cases of the presented model, all of which involve study cases where $\alpha \geq 0.5$. However, they do not discuss cases where $\alpha$ is less than 0.5 or tends towards zero. Therefore, this paper aims to handle this issue by developing and applying an implicit method approach to solve the fuzzy time fractional cancer tumor model. Our proposed method is unconditionally stable, indicating its ability to encompass all scenarios and handle all potential cases specifically focusing on scenarios where the net cell-killing rate is time dependent. Furthermore, we examine the implications of employing fractional derivatives instead of integer derivatives across various fractional order values.

## 2. Modeling cancer tumor dynamics with time-fractional approaches in a fuzzy environment

This section delves into the overarching structure of the FTFCTM as elaborated through the fundamental principles of fuzzy theory and several associated characteristics found in references [31–34].

Consider the general one dimensional FTFTM:

$$\frac{\partial^{\alpha}\tilde{u}(x,t,\alpha)}{\partial^{\alpha}t} = \frac{\partial^{2}\tilde{u}(x,t)}{\partial x^{2}} - \tilde{k}(x,t)\,\tilde{u}(x,t),\ 0 < \alpha \leq 1,\ (x,t)\epsilon\,\Omega = [0,\mathrm{L}] \times [0,\mathrm{T}] \qquad (1)$$

Alongside the boundary and initial conditions

$$\tilde{u}(x,0) = \tilde{f}(x),\ \tilde{u}(0,t) = \tilde{m}(0,t),\ \tilde{u}(l,t) = \tilde{n}(l,t),$$

Here, $\tilde{u}(x,t,\alpha)$ denotes the concentration of tumor cells at fractional order $\alpha$ and time t, The net cell-killing rate, denoted as $\tilde{k}(x,t)$, represents the fuzzy net effect on crisp variables $x$

and $t$. $\frac{\partial^\alpha \tilde{u}(x,t,\alpha)}{\partial^\alpha t}$ is the fuzzy time fractional derivative of order $\alpha$ [35], $\frac{\partial^2 \tilde{U}(x,t)}{\partial x^2}$ indicate the fuzzy Hukuhara derivatives concerning $x$ as well as the fuzzy initial condition represented by $\tilde{u}(0,x)$. The fuzzy boundary conditions are expressed as $\tilde{u}(0,t)$ and $\tilde{u}(l,0)$, corresponding to the fuzzy convex numbers $\tilde{m}$ and $\tilde{n}$, respectively. Furthermore, the fuzzy functions $\tilde{k}(x,t), \tilde{f}(x)$ are defined as follows [36].

$$\begin{cases} \check{k}(x,t) = \tilde{\tau}_1 \, s_1(x,t) \\ \tilde{f}(x) = \tilde{\tau}_2 \, s_2(x) \end{cases}, \tag{2}$$

where $s_1(x,t)$ and $s_2(x)$ are the functions of the variable $x$ and t, while $\tilde{\tau}_1$ and $\tilde{\tau}_2$ represent the fuzzy convex numbers. The defuzzification of FTFCTM involves employing a singular parametric approach for handling fuzzy numbers.

The process of defuzzifying Eq (1) is presented for all values of $r$ within the range of * 0,1] in the following manner [36]:

$$[\tilde{u}(x,t)]_r = \underline{u}(x,t;r), \bar{u}(x,t;r) \tag{3}$$

$$\left[\frac{\partial^\alpha \tilde{u}(x,t,\alpha)}{\partial^\alpha t}\right]_r = \frac{\partial^\alpha \underline{u}(x,t,\alpha;r)}{\partial^\alpha t}, \frac{\partial^\alpha \bar{u}(x,t,\alpha;r)}{\partial^\alpha t} \tag{4}$$

$$\left[\frac{\partial^2 \tilde{u}(x,t)}{\partial x^2}\right]_r = \frac{\partial^2 \underline{u}(x,t;r)}{\partial x^2}, \frac{\partial^2 \bar{u}(x,t;r)}{\partial x^2} \tag{5}$$

$$\left[\tilde{k}(x,t)\right]_r = \underline{k}(x,t;r), \bar{k}(x,t;r) \tag{6}$$

$$[\tilde{u}(x,0)]_r = \underline{u}(x,0;r), \bar{u}(x,0;r) \tag{7}$$

$$[\tilde{u}(0,t)]_r = \underline{u}(0,t;r), \bar{u}(0,t;r) \tag{8}$$

$$[\tilde{u}(l,t)]_r = \underline{u}(l,t;r), \bar{u}(l,t;r) \tag{9}$$

$$\left[\tilde{f}(x)\right]_r = \underline{f}(x;r), \bar{f}(x;r) \tag{10}$$

$$\begin{cases} [\tilde{m}]_r = \underline{m}(t;r), \bar{m}(t;r) \\ [\tilde{n}]_r = \underline{n}(l;r), \bar{n}(l;r) \end{cases} \tag{11}$$

where

$$\begin{cases} \left[\tilde{k}(x,t)\right]_r = \left[\underline{\tau}(r)_1, \bar{\tau}_1(r)\right] s_1(x,t) \\ \left[\tilde{f}_1(x)\right]_r = \left[\underline{\tau}(r)_2, \bar{\tau}_2(r)\right] s_2(x) \end{cases}. \tag{12}$$

The extension principle is employed to define the membership function [36]

$$
\begin{cases}
\underline{u}(x,t;r) = min\{\tilde{u}(\tilde{\mu}(r),t))|\tilde{\mu}(r) \in \tilde{u}(x,t;r)\} \\
\bar{u}(x,t;r) = max\{\tilde{u}(\tilde{\mu}(r),t)|\tilde{\mu}(r) \in \tilde{u}(x,t;r)\}
\end{cases}. \tag{13}
$$

Now Eq (1) for $0 \leq \alpha \leq 1$, $t > 0$ and $r \in [0,1]$ has been reformulated to derive of the FTFTM as follows:

$$
\begin{cases}
\dfrac{\partial^{\alpha}\underline{u}(x,t,\alpha;r)}{\partial^{\alpha}t} = \dfrac{\partial^{2}\underline{u}(x,t;r)}{\partial x^{2}} - \underline{\tau}(r)_{1}\, s_{1}(x,t)\,\underline{u}(x,t;r),\ 0 < \alpha \leq 1,\ (x,t)\epsilon\,\Omega = [0,\text{L}] \times [0,\text{T}] \\
\underline{u}(x,0;r) = \underline{\tau}(r)_{2}\, s_{2}(x) \\
\underline{u}(0,t;r) = \underline{m}(r), \underline{u}(l,t;r) = \underline{n}(r)
\end{cases} \tag{14}
$$

$$
\begin{cases}
\dfrac{\partial^{\alpha}\bar{u}(x,t,\alpha;r)}{\partial^{\alpha}t} = \dfrac{\partial^{2}\bar{u}(x,t;r)}{\partial x^{2}} - \bar{\tau}_{1}(r)\, s_{1}(x,t)\,\bar{u}(x,t;r),\ 0 < \alpha \leq 1,\ (x,t)\epsilon\,\Omega = [0,\text{L}] \times [0,\text{T}] \\
\bar{u}(x,0;r) = \bar{\tau}_{2}(r)\, s_{2}(x) \\
\bar{u}\,(0,t;r) = \bar{m}(r), \bar{u}(l,t;r) = \bar{n}(r)
\end{cases} \tag{15}
$$

The Eqs (14) and (15) presented the upper and lower bounds respectively of the general from of FTFTM.

## 3. Implicit scheme for solving FTFCTM

In this section, an implicit scheme is implemented in Caputo sense for time-fractional derivative and central difference approximation at time level $n + 1$, for second order space derivative, to solve FTFCTM in single parametric form of fuzzy numbers. The time-fractional derivative in Eq (15) is discretized using the Caputo formula as [37,38]

$$
\frac{\partial^{\alpha}\tilde{u}(x,t,\alpha)}{\partial^{\alpha}t} =
\begin{cases}
\dfrac{\Delta t^{-\alpha}}{\Gamma(2-\alpha)}\left[\underline{u}_{i}^{n+1}(x,t;r) - \underline{u}_{i}^{n}(x,t;r) + \sum_{j=1}^{n} b_{j}(\underline{u}_{i}^{n+1-j}(x,t;r) - \underline{u}_{i}^{n-j}(x,t;r))\right] + O(\Delta t) \\
\dfrac{\Delta t^{-\alpha}}{\Gamma(2-\alpha)}\left[\bar{u}_{i}^{n+1}(x,t;r) - \bar{u}_{i}^{n}(x,t;r) + \sum_{j=1}^{n} b_{j}(\bar{u}_{i}^{n+1-j}(x,t;r) - \bar{u}_{i}^{n-j}(x,t;r))\right] + O(\Delta t),
\end{cases} \tag{16}
$$

where $b_{j} = (j + 1)^{1-\alpha} - (j)^{1-\alpha}$, $j = 1,2,\ldots.$

The central difference approximation at time level $n + 1$ is used to discretize the second partial derivative as follows

$$
\frac{\partial^{2}\tilde{u}(x,t;r)}{\partial x^{2}} =
\begin{cases}
\dfrac{\underline{u}_{i+1}^{n+1}(x,t;r) - 2\underline{u}_{i}^{n+1}(x,t;r) + \underline{u}_{i-1}^{n+1}(x,t;r)}{\Delta x^{2}} \\
\dfrac{\bar{u}_{i+1}^{n+1}(x,t;r) - 2\bar{u}_{i}^{n+1}(x,t;r) + \bar{u}_{i-1}^{n+1}(x,t;r)}{\Delta x^{2}}
\end{cases} \tag{17}
$$

We substitute Eqs (16) and (17) into Eq (15) to obtain

$$\frac{\Delta t^{-\alpha}}{\Gamma(2-\alpha)}\left[\underline{u}_i^{n+1}(x,t;r) - \underline{u}_i^n(x,t;r) + \sum_{j=1}^n b_j\left(\underline{u}_i^{n+1-j}(x,t;r) - \underline{u}_i^{n-j}(x,t;r)\right)\right]$$
$$= \frac{\underline{u}_{i+1}^{n+1}(x,t;r) - 2\underline{u}_i^{n+1}(x,t;r) + \underline{u}_{i-1}^{n+1}(x,t;r)}{\Delta x^2} - \underline{k}(x,t;r)\,\underline{u}_i^n(x,t;r). \qquad (18)$$

$$\frac{\Delta t^{-\alpha}}{\Gamma(2-\alpha)}\left[\bar{u}_i^{n+1}(x,t;r) - \bar{u}_i^n(x,t;r) + \sum_{j=1}^n b_j\left(\bar{u}_i^{n+1-j}(x,t;r) - \bar{u}_i^{n-j}(x,t;r)\right)\right]$$
$$= \frac{\bar{u}_{i+1}^{n+1}(x,t;r) - 2\bar{u}_i^{n+1}(x,t;r) + \bar{u}_{i-1}^{n+1}(x,t;r)}{\Delta x^2} - \bar{k}(x,t;r)\bar{u}_i^n(x,t;r). \qquad (19)$$

By letting $\tilde{\tau}(r) = \frac{\Delta t^\alpha\,\Gamma(2-\alpha)}{\Delta x^2}$ and $r \in [0,1]$, and employing Eqs (18) and (19) respectively, we get

$$-\tilde{\tau}(r)\left(\underline{u}_{i+1}^{n+1}(x,t;r) + \underline{u}_{i-1}^{n+1}(x,t;r)\right) + (1+2\tau)\,\underline{u}_i^{n+1}(x,t;r)$$
$$= (1 - \Delta t^\alpha\Gamma(2-\alpha)\underline{k}(x,t;r))\underline{u}_i^n(x,t;r) - \sum_{j=1}^n b_j\left(\underline{u}_i^{n+1-j}(x,t;r) - \underline{u}_i^{n-j}(x,t;r)\right). \qquad (20)$$

$$-\tilde{\tau}(r)\left(\bar{u}_{i+1}^{n+1}(x,t;r) + \bar{u}_{i-1}^{n+1}(x,t;r)\right) + (1+2\tau)\bar{u}_i^{n+1}(x,t;r)$$
$$= \left(1 - \Delta t^\alpha\Gamma(2-\alpha)\bar{k}(x,t;r)\right)\bar{u}_i^n(x,t;r) - \sum_{j=1}^n b_j\left(\bar{u}_i^{n+1-j}(x,t;r) - \bar{u}_i^{n-j}(x,t;r)\right). \qquad (21)$$

## 4. Stability analysis

It is first assumed that the discretization of the initial condition yields the fuzzy error $\tilde{\varepsilon}_i^0$. Let $\tilde{u}_i^0 = \acute{\tilde{u}}_i^0 - \tilde{\varepsilon}_i^0$, $\tilde{u}_i^n$ and $\acute{\tilde{u}}_i^n$ be the fuzzy numerical solutions of the scheme of the Eqs (20) and (21), with respect to the initial data's $\tilde{m}_i^0$ and $\acute{\tilde{n}}_i^0$, respectively. Let $[\tilde{u}_i^n(x,t;\alpha)]_r = [\underline{u}_i^n(r), \bar{u}_i^n(r)]$, where $r \in [0,1]$. Then, the fuzzy error bound is defined as $\left[\tilde{\varepsilon}_i^n\right]_r = \left[\acute{\tilde{u}}_i^n - \tilde{u}_i^n\right]_r$ where:

$$\left[\tilde{\varepsilon}_i^n\right]_r = \left\{\underline{\varepsilon}i^n(r), \bar{\varepsilon}_i^n(r)\right\} = \begin{cases} \underline{u}_i^{n\prime}(r) - \underline{u}_i^n(r) \\ \bar{u}_i^{n\prime}(r) - \bar{u}_i^n(r) \end{cases}, \; n = 1, 2, \;\ldots\ldots N-1, \; i = 1, 2, \ldots, M-1. \quad (22)$$

Presently, in accordance with the methodology employed in reference [39], the Eqs (20) and (21) can be rewritten as follows

$$\begin{cases} -\tau\,\underline{u}_{i+1}^{n+1} + (1+2\tau)\underline{u}_i^{n+1} - \tau\,\underline{u}_{i-1}^{n+1} = (1 - \Delta t^\alpha\,\Gamma(2-\alpha)\,\tilde{k}(x,t) - b_1)\underline{u}_i^n - \sum_{j=1}^{n-1}\left(b_{j+1} - b_j\right)\underline{u}_i^{n-j} + b_n\underline{u}_i^0 \\ -\tau\,\bar{u}_{i+1}^{n+1} + (1+2\tau)\bar{u}_i^{n+1} - \tau\,u_{i-1}^{n+1} = (1 - \Delta t^\alpha\,\Gamma(2-\alpha)\,\tilde{k}(x,t) - b_1)\bar{u}_i^n - \sum_{j=1}^{n-1}\left(b_{j+1} - b_j\right)\bar{u}_i^{n-j} + b_n\bar{u}_i^0 \end{cases} \quad (23)$$

Since we assume $\tilde{\tau}(r) = \frac{\Delta t^\alpha\,\Gamma(2-\alpha)}{\Delta x^2}$ so, the $\Delta t^\alpha\,\Gamma(2-\alpha)\,\tilde{k}(x,t) = \tau\,\Delta x^2\,\tilde{k}(x,t)$. Hence, we rephrase the imprecise rounding error associated with Eq (23) in the following manner:

$$\begin{cases} -\tau\,\underline{\varepsilon}_{i+1}^{n+1} + (1+2\tau)\underline{\varepsilon}_i^{n+1} - \tau\,\underline{\varepsilon}_{i-1}^{n+1} = \left(1 - \tau\,\Delta x^2\,\tilde{k}(x,t) - b_1\right)\underline{\varepsilon}_i^n - \sum_{j=1}^{n-1}\left(b_{j+1} - b_j\right)\underline{\varepsilon}_i^{n-j} + b_n\underline{\varepsilon}_i^0 \\ -\tau\,\bar{\varepsilon}_{i+1}^{n+1} + (1+2\tau)\bar{\varepsilon}_i^{n+1} - \tau\,\bar{\varepsilon}_{i-1}^{n+1} = \left(1 - \tau\,\Delta x^2\,\tilde{k}(x,t) - b_1\right)\bar{\varepsilon}_i^n - \sum_{j=1}^{n-1}\left(b_{j+1} - b_j\right)\bar{\varepsilon}_i^{n-j} + b_n\bar{\varepsilon}_i^0 \end{cases} \quad (24)$$

Assume $\tilde{\varepsilon}_0^n = \tilde{\varepsilon}_X^n = 0$, $n = 1, 2, \ldots, N-1$ and $\tilde{\varepsilon}_i^n = [\tilde{\varepsilon}_1^n, \tilde{\varepsilon}_2^n, \ldots\ldots, \tilde{\varepsilon}_{X-1}^n]$. Then, introduce the fuzzy norm

$$\|\tilde{\varepsilon}^n\|_2 = \sqrt{\sum_{i=1}^{X-1} h \, |\tilde{\varepsilon}_i^n|^2}, \tag{25}$$

which gives

$$\|\tilde{\varepsilon}^n\|_2^2 = \sum_{i=-\infty}^{\infty} |\tilde{\lambda}^n|^2. \tag{26}$$

Hence, $\tilde{\varepsilon}_i^n$ may alternatively be expressed as

$$\begin{cases} \underline{\varepsilon}_i^n = \underline{\lambda}^n \, e^{\sqrt{-\theta_i}} \\ \bar{\varepsilon}_i^n = \bar{\lambda}^n \, e^{\sqrt{-\theta_i}} \end{cases}, \text{ where } \tilde{\theta}_i = qih. \tag{27}$$

Therefore, by substituting Eqs (27) into (24), we derive:

$$-\tau \, \underline{\lambda}^{n+1} \, e^{\sqrt{-\theta_{i+1}}} + (1 + 2\tau)\underline{\lambda}^{n+1} \, e^{\sqrt{-\theta_i}} - \tau \, \underline{\lambda}^{n+1} \, e^{\sqrt{-\theta_{i-1}}}$$
$$= (1 - \tau \, \Delta x^2 \, \tilde{k}(x,t) - b_1)\underline{\lambda}^n \, e^{\sqrt{-\theta_i}} - \sum_{j=1}^{n-1} \left(b_{j+1} - b_j\right)\underline{\lambda}^{n-j} \, e^{\sqrt{-\theta_i}} + b_n\underline{\lambda}^0 \, e^{\sqrt{-\theta_i}} \tag{28}$$

$$-\tau \, \bar{\lambda}^{n+1} \, e^{\sqrt{-\theta_{i+1}}} + (1 + 2\tau)\bar{\lambda}^{n+1} \, e^{\sqrt{-\theta_i}} - \tau \, \bar{\lambda}^{n+1} \, e^{\sqrt{-\theta_{i-1}}}$$
$$= (1 - \tau \, \Delta x^2 \, \tilde{k}(x,t) - b_1)\bar{\lambda}^n \, e^{\sqrt{-\theta_i}} - \sum_{j=1}^{n-1} \left(b_{j+1} - b_j\right)\bar{\lambda}^{n-j} \, e^{\sqrt{-\theta_i}} + b_n\bar{\lambda}^0 \, e^{\sqrt{-\theta_i}} \tag{29}$$

Dividing Eqs (28) and (29) by $e^{\sqrt{-\theta_i}}$ reveals:

$$\begin{cases} [1 + 2\tau - \tau \, (e^{\sqrt{-\theta_i}} + e^{-\sqrt{-\theta_i}})] \, \underline{\lambda}^{n+1} = (1 - \tau \, \Delta x^2 \, \tilde{k}(x,t) - b_1)\underline{\lambda}^n - \sum_{j=1}^{n-1} \left(b_{j+1} - b_j\right)\underline{\lambda}^{n-j} + b_n\underline{\lambda}^0 \\ [1 + 2\tau - \tau \, (e^{\sqrt{-\theta_i}} + e^{-\sqrt{-\theta_i}})] \, \bar{\lambda}^{n+1} = (1 - \tau \, \Delta x^2 \, \tilde{k}(x,t) - b_1)\bar{\lambda}^n - \sum_{j=1}^{n-1} \left(b_{j+1} - b_j\right)\bar{\lambda}^{n-j} + b_n\bar{\lambda}^0 \end{cases} \tag{30}$$

By simplifying the Eq (30) we get:

$$\begin{cases} \underline{\lambda}^{n+1} = \left(\dfrac{\left(1 - \tau \, \Delta x^2 \, \tilde{k}(x,t) - b_1\right)}{1 + 4p\sin^2\left(\dfrac{\theta}{2}\right)}\right) \underline{\lambda}^n + \dfrac{\sum_{j=1}^{n-1} \left(b_{j+1} - b_j\right)\underline{\lambda}^{n-j} + b_n\underline{\lambda}^0}{1 + 4p\sin^2\left(\dfrac{\theta}{2}\right)} \\[4mm] \bar{\lambda}^{n+1} = \left(\dfrac{\left(1 - \tau \, \Delta x^2 \, \tilde{k}(x,t) - b_1\right)}{1 + 4p\sin^2\left(\dfrac{\theta}{2}\right)}\right) \bar{\lambda}^n + \dfrac{\sum_{j=1}^{n-1} \left(b_{j+1} - b_j\right)\bar{\lambda}^{n-j} + b_n\bar{\lambda}^0}{1 + 4p\sin^2\left(\dfrac{\theta}{2}\right)} \end{cases} \tag{31}$$

**Proposition 1:** if $\tilde{\lambda}^n$ is the fuzzy solution of the Eq (31) then $\begin{cases} |\underline{\lambda}^n| \le |\underline{\lambda}^0| \\ |\bar{\lambda}^n| \le |\bar{\lambda}^0| \end{cases}$

**Proof:**

From Eq (31) and when $n = 0$, we obtain:

$$\begin{cases} \underline{\lambda}^1 = \left( \dfrac{1}{1 + 4\tau\sin^2\left(\frac{\theta}{2}\right)} \right)\underline{\lambda}^0 \\ \bar{\lambda}^1 = \left( \dfrac{1}{1 + 4\tau\sin^2\left(\frac{\theta}{2}\right)} \right)\bar{\lambda}^0 \end{cases}$$

Since $4\tau\sin^2\left(\frac{\theta}{2}\right) \geq 0$, we have:

$$\begin{cases} |\underline{\lambda}^1| \leq |\tilde{\lambda}^0| \\ |\bar{\lambda}^n| \leq |\bar{\lambda}^0| \end{cases}$$

Now suppose that

$$\begin{cases} |\underline{\lambda}^m| \leq |\tilde{\lambda}^0| \\ |\bar{\lambda}^m| \leq |\bar{\lambda}^0| \end{cases}, \; m = 1,\, 2,\, 3,\, \ldots,\, n-1$$

**Lemma 1:** the coefficients $b_j = (j+1)^{1-\alpha} - (j)^{1-\alpha}, j = 1,2,\ldots$ satisfy (Chen et al., 2007) [40]:

1. $0 < b_j \leq 1, \mathrm{j} = 1, 2, 3,\ldots$

2. $b_j > b_{j+1}\, j = 1,2,3,\ldots$

3. $\sum_{j=1}^{n-1}\left(b_{j+1} - b_j\right) = 1 - b_n$

From lemma 1 and Eq (31) we obtain:

$$\begin{cases} |\underline{\lambda}^{n+1}| \leq \dfrac{[(1 - b_1) - (b_n - b_1) + b_n]}{1 + 4p\sin^2\left(\dfrac{\theta}{2}\right)}|\underline{\lambda}^0| \leq |\underline{\lambda}^0| \\[4mm] |\bar{\lambda}^{n+1}| \leq \dfrac{[(1 - b_1) - (b_n - b_1) + b_n]}{1 + 4p\sin^2\left(\dfrac{\theta}{2}\right)}|\bar{\lambda}^0| \leq |\bar{\lambda}^0| \end{cases}$$

**Theorem 1:** The fuzzy lower and fuzzy upper implicit Scheme in Eqs (20) and (21) respectivley is unconditionally stable

**Proof.** From the formula in Eq (26) and proposition 1, it can be obtained that

$$\|\tilde{\varepsilon}^n\|_2 \leq \|\tilde{\varepsilon}^0\|_2, \; n = 1, 2, \ldots, N - 1$$

which means fuzzy lower and fuzzy upper implicit Scheme in Eqs (20) and (21) is unconditionally stable.

## 5. Numerical experiment and discussion

Consider the FTFCTM when the net cell-killing rate is only time dependent [4]

$$\frac{\partial^\alpha \tilde{u}(x, t, \alpha)}{\partial^\alpha t} = \frac{\partial^2 \tilde{u}(x, t)}{\partial x^2} - t^2\, \tilde{u}(x, t), \; 0 < \alpha \leq 1. \tag{32}$$

Let's consider the uncertain initial condition is $\tilde{u}(x, 0) = \tilde{s}(r)\, e^{kx}$, where

$$\tilde{s}(r) = [0.75 + 0.25\, r,\, 1.25 - 0.25r],\; r \in [0, 1].$$

where, the fuzzy analytical solution of Eq (32) is obtained in [4] as follows:

$$\tilde{u}(x, t, \alpha) = \tilde{s}(r)\left( e^{kx} + e^{kx}k^2 \, \frac{t^\alpha}{\Gamma(1+\alpha)} + \frac{t^{2\alpha}}{\Gamma(1+2\alpha)} \, e^{kx}k^4 \right).$$

Hence, the definition of the absolute error for the obtained solution of Eq (32) can be expressed as:

$$\left[\tilde{E}\right]_r = |\tilde{U}(t, x; r) - \tilde{u}(t, x; r)|.$$

To solve the FTFCTM in Eq (32), we implement the implicit method discussed in Section 3 to obtain a tridiagonal system of equations, where each equation involves the unknowns at the current and neighboring grid points. The obtained tridiagonal system of linear equations typically requires the use of iterative methods for solving sparse linear systems. In each time step, we solve a linear system of equations using Wolfram Mathematica 11.2 to obtain the values $\tilde{u}(x, t, \alpha)$ for that particular time level.

At $\Delta t = 0.01, \Delta x = 0.5$ and, $\tilde{\tau}(r) = \frac{\Delta t^\alpha \, \Gamma(2-\alpha)}{\Delta x^2}$, we have the following results:

Table 1, Figs 1A, 1B, 2A, 2B, 3A, 3B, 4A and 4B demonstrate that the implicit finite difference scheme exhibits strong agreement with the exact solution and adheres to the characteristics of triangular fuzzy number shapes. Moreover, the results presented in Table 1, Figs 1A, 1B, 2A and 2B clearly indicate that the numerical outcomes become increasingly precise as α approaches 1. Furthermore, the precision of the fuzzy numerical solution is contingent on the value of r, with fuzzy lower solution accuracy improving as r decreases, while the opposite holds true for the fuzzy upper solution. Figs 3A, 3B and 4A, depict the three-dimensional graphical representations of the fuzzy implicit method and the analytical solution. These figures reveal that the rate of cancer cell reduction increases over time, thereby validating the proposed approach across various time and space intervals while adhering to the stability conditions discussed in Section 4.

**Table 1. Fuzzy numerical solution of Eq (32), by implicit scheme at different values of $\alpha$ and $r$ at $t = 0.05$, $x = 4$.**

| | | Fuzzy Lower solution | | Fuzzy Upper solution | |
|---|---|---|---|---|---|
| $r$ | $\alpha$ | $\underline{u}(4, 0.05; r, \alpha)$ | $\underline{E}(4, 0.05; r, \alpha)$ | $\bar{u}(4, 0.05; r, \alpha)$ | $\bar{E}(4, 0.05; r, \alpha)$ |
| $r = 0$ | 0.2 | 0.02070978 | $1.11503 \times 10^{-2}$ | 0.03451629 | $1.85838 \times 10^{-2}$ |
| | 0.4 | 0.01806187 | $2.13199 \times 10^{-3}$ | 0.0301031 | $3.55332 \times 10^{-3}$ |
| | 0.6 | 0.01592767 | $7.40335 \times 10^{-4}$ | 0.02654611 | $1.23389 \times 10^{-3}$ |
| | 0.8 | 0.01487763 | $2.84872 \times 10^{-4}$ | 0.02479604 | $4.74786 \times 10^{-4}$ |
| | 1 | 0.01437086 | $7.01618 \times 10^{-5}$ | 0.0239514 | $1.16936 \times 10^{-4}$ |
| | | | | | |
| $r = 0.3$ | 0.2 | 0.02347108 | $1.1575 \times 10^{-2}$ | 0.03244532 | $1.74688 \times 10^{-2}$ |
| | 0.4 | 0.01986805 | $2.34519 \times 10^{-3}$ | 0.0282969 | $3.34012 \times 10^{-3}$ |
| | 0.6 | 0.01752043 | $8.14369 \times 10^{-4}$ | 0.02495335 | $1.15986 \times 10^{-3}$ |
| | 0.8 | 0.01636539 | $3.13359 \times 10^{-4}$ | 0.02330828 | $4.46299 \times 10^{-4}$ |
| | 1 | 0.01580795 | $7.7178 \times 10^{-5}$ | 0.02251435 | $1.0992 \times 10^{-4}$ |
| | | | | | |
| $r = 0.7$ | 0.2 | 0.02554206 | $1.3752 \times 10^{-2}$ | 0.02968401 | $1.59821 \times 10^{-2}$ |
| | 0.4 | 0.02227630 | $2.62946 \times 10^{-3}$ | 0.02588868 | $3.05585 \times 10^{-3}$ |
| | 0.6 | 0.019644127 | $9.1308 \times 10^{-4}$ | 0.02282966 | $1.06115 \times 10^{-3}$ |
| | 0.8 | 0.01834907 | $3.51342 \times 10^{-4}$ | 0.02132460 | $4.08316 \times 10^{-4}$ |
| | 1 | 0.01772406 | $8.65329 \times 10^{-5}$ | 0.02059824 | $1.00565 \times 10^{-4}$ |

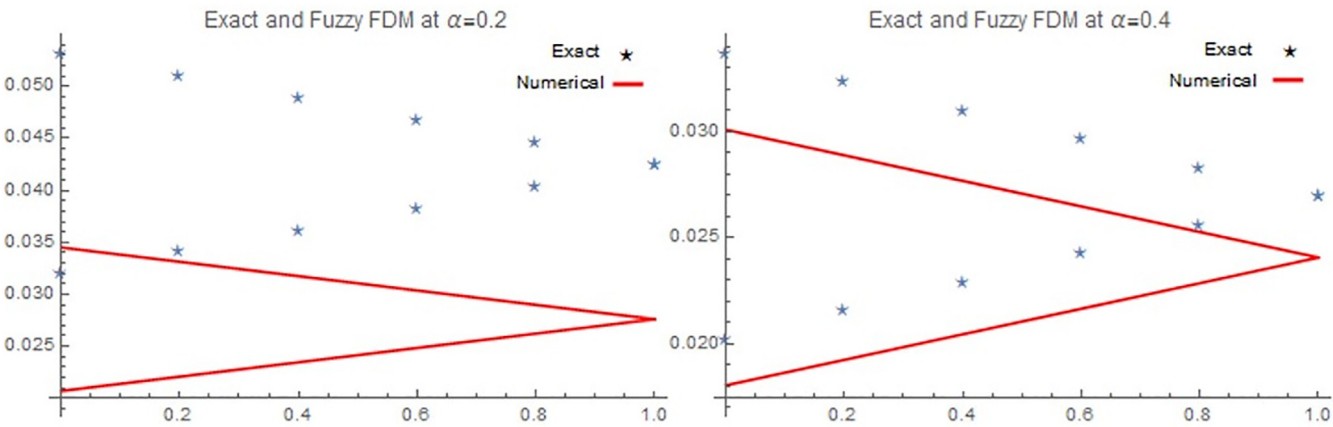

**Fig 1. (A)** Numerical and Exact solution of Eq (32), by implicit scheme at $\alpha = 0.2$, **(B)** Numerical and Exact solution of Eq (32) at $\alpha = 0.4$ for $t = 0.05$, $x = 4$, $\alpha$ for all $r \in [0,1]$.

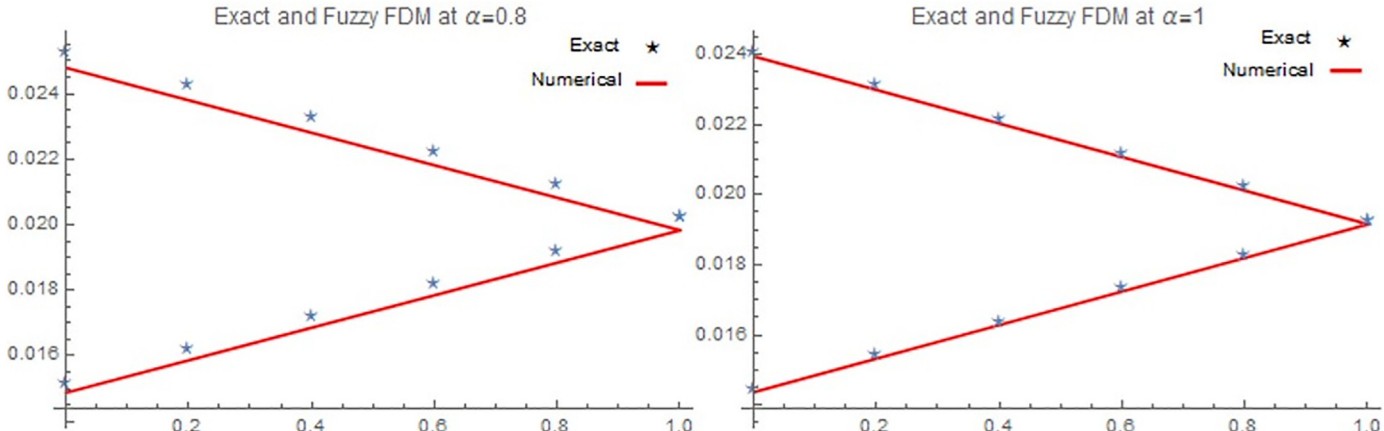

**Fig 2. (A)** Numerical and Exact solution of Eq (32), by implicit scheme at $\alpha = 0.8$, **(B)** Numerical and Exact solution of Eq (32) at $\alpha = 1$ for $t = 0.05$, $x = 4$, $\alpha$ for all $r \in [0,1]$.

From the information presented above, it is evident that employing the FTFCTM is more effective when compared to the classical fractional cancer model. This is due to its enhanced precision in predicting cancer tumor spread and growth, as well as its capacity to handle uncertainties and ambiguities in data, like as the uncertainties related to the initial condition in Eq (32) as demonstrated in the provided example. Additionally, the fuzzy fractional model can effectively capture non-linear and non-instantaneous behaviors in tumor growth

## 6. Conclusions

This paper examines the influence of substituting a fuzzy time-fractional derivative for the conventional time derivative within the context of the fuzzy cancer tumor model. The investigation takes into consideration various fractional derivative values across multiple scenarios involving fuzzy initial conditions for the FTFCTM. We have proposed an implicit finite

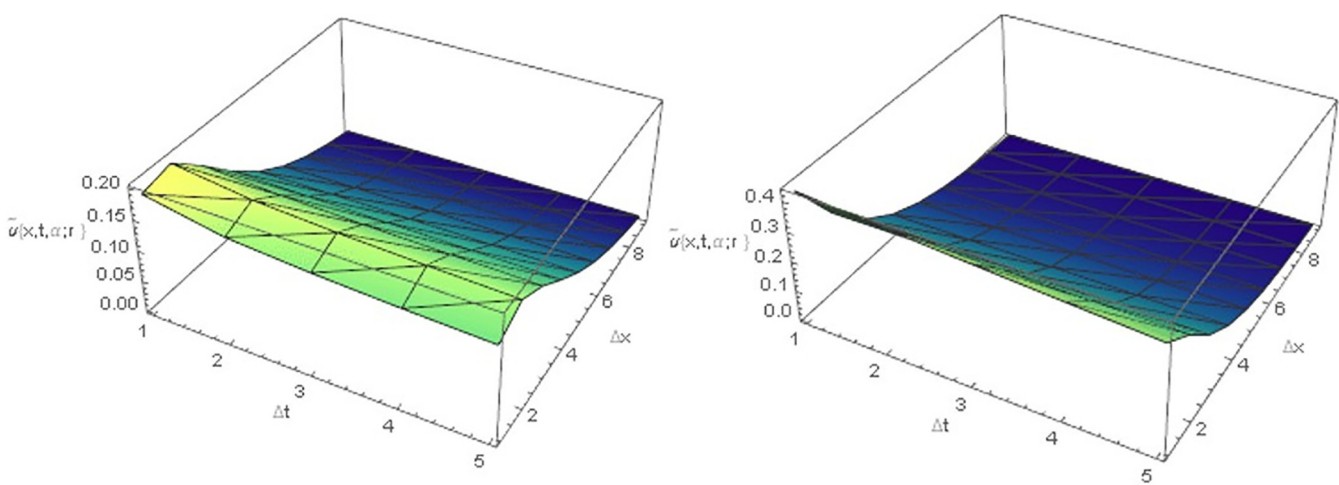

**Fig 3. (A) Lower fuzzy numerical solution of Eq (32) at $\alpha$ = 0.2 (B) Lower fuzzy numerical solution of Eq (32) at $\alpha$ = 1 for all $r \in [0,1]$.**

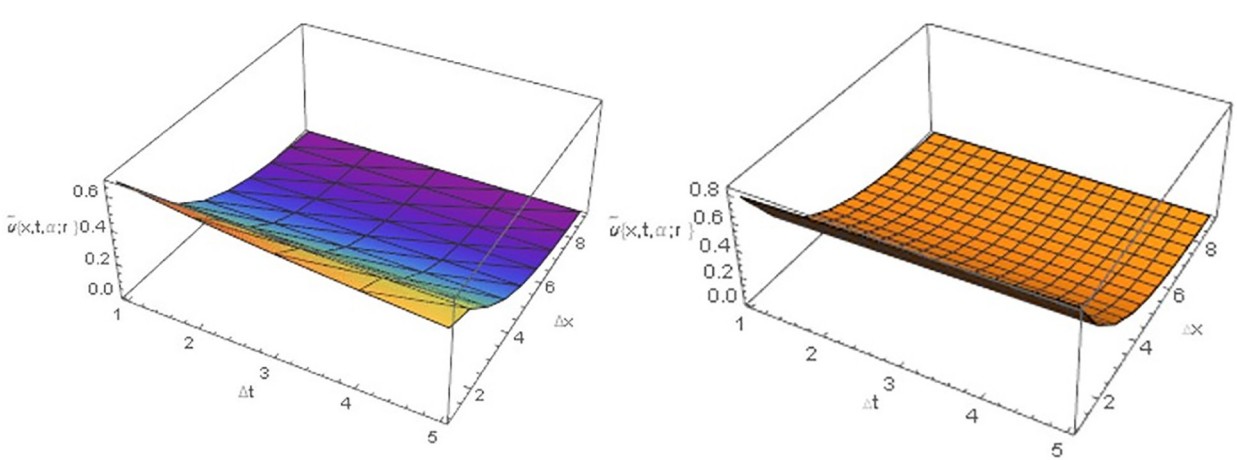

**Fig 4. (A) upper fuzzy numerical solution of Eq (32) at $\alpha$ = 0.8 (B) upper fuzzy exact solution for all $r \in [0,1]$.**

difference technique to tackle the numerical solution of the FTFCTM. In particular, our attention is directed toward scenarios where the net cell-killing rate is solely contingent on time. Caputo's definition was used to replace the time-fractional derivative, and the von Neumann method was employed to assess the stability of the presented numerical scheme. Finally, a numerical illustration has been provided to assess the viability of the suggested method and to verify specific associated facets. The discovery highlights a significant demand for investigating the fuzzy fractional cancer tumor model. This model offers a holistic perspective on cancer tumor behavior, particularly by encompassing various fuzzy scenarios in the initial conditions. Such insight could aid researchers in selecting specific treatment approaches. Furthermore, there is potential for expanding this approach to investigate the correlation between the fuzzy fractional cancer model and the bifurcation analysis of fractional tumor models, a topic that will be thoroughly explored in the future.

## Supporting information

**S1 File.**

(NB)

**S2 File.**

(NB)

**S3 File.**

(NB)

## Author Contributions

**Conceptualization:** Hamzeh Zureigat, Saleh Alshammari.

**Formal analysis:** Hamzeh Zureigat.

**Investigation:** M. Mossa Al-Sawallah.

**Resources:** Mohammad Alshammari.

**Software:** Hamzeh Zureigat.

**Supervision:** M. Mossa Al-Sawallah.

**Validation:** Saleh Alshammari, Mohammed Al-Smadi.

**Writing – review & editing:** Mohammed Al-Smadi.

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
