## [Decision Letter · Decision Letter 0]

22 Mar 2024

PONE-D-24-06051An In-Depth Examination of the Fuzzy Fractional Cancer Tumor Model and Its Numerical Solution by Implicit Finite Difference MethodPLOS ONE

Dear Dr. Zureigat,

Thank you for submitting your manuscript to PLOS ONE. After careful consideration, we feel that it has merit but does not fully meet PLOS ONE’s publication criteria as it currently stands. Therefore, we invite you to submit a revised version of the manuscript that addresses the points raised during the review process.

We look forward to receiving your revised manuscript.

Kind regards,

Viacheslav Kovtun, Dr.Sc., Ph.D.

Academic Editor

PLOS ONE

Journal Requirements:

"This research has been funded by scientific research deanship at university of Ha’il -Saudi Arabia through project numemper (RG-23122)."

"This research has been funded by scientific research deanship at university of Ha’il -Saudi Arabia through

project numemper (RG-23122)."

Please remove any funding-related text from the manuscript. 

5. Thank you for stating the following in your Competing Interests section: "NO authors have competing interests"

6. We note that your Data Availability Statement is currently as follows: "All relevant data are within the manuscript and its Supporting Information files."

Reviewers' comments:

Reviewer's Responses to Questions

**Comments to the Author**

1. Is the manuscript technically sound, and do the data support the conclusions?

Reviewer #1: Yes

Reviewer #2: Yes

2. Has the statistical analysis been performed appropriately and rigorously? 

Reviewer #1: Yes

Reviewer #2: Yes

3. Have the authors made all data underlying the findings in their manuscript fully available?

Reviewer #1: Yes

Reviewer #2: Yes

4. Is the manuscript presented in an intelligible fashion and written in standard English?

Reviewer #1: Yes

Reviewer #2: No

5. Review Comments to the Author

Reviewer #1: Dear Author, I have reviewed your manuscript entitled "An In-Depth Examination of the Fuzzy Fractional Cancer Tumor Model and Its Numerical Solution by Implicit Finite Difference Method" and I found that it is an interesting study by solving fuzzy fractional PDE using an implicit FDM. However, it needs minor revisions before it can be published in the journal PLOS ONE. Below are my comments:

Abstract:

1. Summarize the main research question and key findings.

Introduction:

1. Elaborate and highlight the main research question.

2. Justify why implicit FDM is used for this study.

3. Remove the displayed Equation 1 if this equation is not used anywhere in the manuscript.

Numerical Experiment and Discussion:

1. Provide one more mathematical problem to illustrate the efficacy of the proposed numerical approach or use Equation 1.

2. Describe the numerical experiment setting such as the software used and the evaluation metric.

Reviewer #2: Dear Editor I read this paper carefully, In my opinion, the novelty is very limited see this published article: 10.3390/ijerph20043766

I think it has the same idea. Also, I have these questions for the authors.

1. What parts do you consider original or relevant for the field? What

specific gap in the field does the paper address?

2. What does this paper add to the subject area compared with other published material such as this article, 10.3390/ijerph20043766?

6. PLOS authors have the option to publish the peer review history of their article (what does this mean?). If published, this will include your full peer review and any attached files.

Reviewer #1: No

Reviewer #2: No

---

## [Decision Letter · Decision Letter 1]

3 May 2024

An In-Depth Examination of the Fuzzy Fractional Cancer Tumor Model and Its Numerical Solution by Implicit Finite Difference Method

PONE-D-24-06051R1

Dear Dr. Zureigat,

We’re pleased to inform you that your manuscript has been judged scientifically suitable for publication and will be formally accepted for publication once it meets all outstanding technical requirements.

Kind regards,

Viacheslav Kovtun, Dr.Sc., Ph.D.

Academic Editor

PLOS ONE

Additional Editor Comments (optional):

Reviewers' comments:

Reviewer's Responses to Questions

**Comments to the Author**

1. If the authors have adequately addressed your comments raised in a previous round of review and you feel that this manuscript is now acceptable for publication, you may indicate that here to bypass the “Comments to the Author” section, enter your conflict of interest statement in the “Confidential to Editor” section, and submit your "Accept" recommendation.

Reviewer #1: All comments have been addressed

2. Is the manuscript technically sound, and do the data support the conclusions?

Reviewer #1: Yes

3. Has the statistical analysis been performed appropriately and rigorously? 

Reviewer #1: Yes

4. Have the authors made all data underlying the findings in their manuscript fully available?

Reviewer #1: Yes

5. Is the manuscript presented in an intelligible fashion and written in standard English?

Reviewer #1: Yes

6. Review Comments to the Author

Reviewer #1: (No Response)

7. PLOS authors have the option to publish the peer review history of their article (what does this mean?). If published, this will include your full peer review and any attached files.

Reviewer #1: No

---

## [Editor Report · Acceptance letter]

15 Aug 2024

PONE-D-24-06051R1 

PLOS ONE

Dear Dr. Zureigat, 

I'm pleased to inform you that your manuscript has been deemed suitable for publication in PLOS ONE. Congratulations! Your manuscript is now being handed over to our production team.

Kind regards, 

on behalf of

Prof. Viacheslav Kovtun 

Academic Editor

PLOS ONE